# Effects of Restricted Feeding on Growth Performance, Intestinal Immunity, and Skeletal Muscle Development in New Zealand Rabbits

**DOI:** 10.3390/ani12020160

**Published:** 2022-01-10

**Authors:** Junyi Zhuang, Tong Zhou, Shaocheng Bai, Bohao Zhao, Xinsheng Wu, Yang Chen

**Affiliations:** College of Animal Science and Technology, Yangzhou University, Yangzhou 225009, China; zjy15162128392@163.com (J.Z.); hyojoo18352766127@126.com (T.Z.); bsc305972572@163.com (S.B.); zhao598841633@163.com (B.Z.); xswu@yzu.edu.cn (X.W.)

**Keywords:** restricted feeding, New Zealand rabbits, growth performance, intestinal immunity, skeletal muscle development, PI3K/Akt signaling pathway

## Abstract

**Simple Summary:**

The high prevalence of gastrointestinal diseases in young rabbits is the major cause of impediment in the development of the rabbit industry. Presently, few companies have adopted methods of restricting feeding to improve the survival rate independent of the effect on their growth and development. To explore the effects of different feeding-restriction levels on the growth performance, intestinal immunity, and skeletal muscle development of meat rabbits, 198 New Zealand meat rabbits of 35 days old were selected and randomly divided into three groups: (1) a control group, (2) a 15% feeding restriction group, and (3) a 30% feeding restriction group, with 66 in each group with an equal number of males and females. The growth performance measurement and health-risk assessment indicators, measurement of digestive enzyme activity, immune and antioxidant indexes, and regulation mechanism were evaluated and explored. Finally, we found that a 30% feeding limit affected the growth and development of skeletal muscle in growing rabbits by regulating the PI3K/Akt signaling pathway.

**Abstract:**

This study aimed to explore the effects of different feeding restriction levels on the growth performance, intestinal immunity, and skeletal muscle development of meat rabbits. Additionally, we studied whether complete compensatory growth could be obtained post 2 weeks of restricted feeding, in order to seek a scientific mode of feeding restriction. Each of three groups was exposed to 3 weeks of feeding restriction and 2 weeks of compensatory growth. The 15% feeding restriction showed a negligible effect on the final body-weight of the rabbits (*p* > 0.05), but significantly reduced the feed-to-weight ratio (*p* < 0.05); reduced diarrhea and mortality; and increased digestive enzyme activity and antioxidant capacity. However, a 30% feeding-restriction level substantially reduced the growth rate of the rabbits (*p* < 0.05), impaired skeletal muscle development, and showed no compensatory growth after 2 weeks of nutritional recovery. Additionally, immunoglobulin and antioxidant enzyme synthesis were impaired due to reduced nutritional levels, and levels of pro-inflammatory factors were increased during the compensation period. The IGF1 mRNA expression decreased significantly (*p* < 0.05), whereas MSTN and FOXO1 expression increased noticeably (*p* < 0.05). Moreover, protein levels of p-Akt and p-p70 decreased significantly in the 15% feeding restriction group. Overall, the 15% feeding limit unaffected the weight and skeletal muscle development of rabbits, whereas the 30% feeding limit affected the growth and development of skeletal muscle in growing rabbits. The PI3K/Akt signaling pathway is plausibly a mediator of this process.

## 1. Introduction

Restricted feeding is the approach to limit the calorie intake to attain particular digestive and physiological characteristics depending on the growth and development of an individual [1]. Though ad libitum feeding is usually applied during rabbit rearing, it has been the major cause of feed waste and a high incidence of digestive diseases and mortality [2]. Regardless of retaining the advantages such as reduced diarrhea and low mortality in young rabbits, the excessive nutritional restriction can also lead to reduced growth and low performance of rabbits [3]. Congruently, one study showed that feed restriction in Hubbard broilers reduced average daily feed intake and had a considerably lower feed-to-weight ratio than the control group [4]. Broiler chickens of 8 and 14 days of age, when subjected to compensation of energy restriction, showed similar levels of average daily gain(ADG), average daily feed intake(ADFI), and feed/gain(F/G) to that with the control group [5]. Concomitantly, it is suggested that animals undergoing restriction can negate possible adverse effects on growth performance by providing nutrition.

The digestion and absorption of nutrients by the animals are mainly dependent on the digestive system and the level of nutrition, which further determine their growth and development [6]. The use of 80–90% free-feeding restrictions throughout the production cycle of rabbits has been reported by Bovera, F et al. to produce rabbits with the same live weight at slaughter age [7]. The effect of quantitatively and linearly reducing feed-intake levels (100-60%) on digestive health and growth in rabbits was reported by Gidenne, T. et al. During periods of restricted feeding, mortality and morbidity in rabbits were significantly reduced from 80% and 70% of the feeding level, respectively [8]. Restriction on feed intake of 7–14-day-old broilers manifested a significant increase in several intestinal enzymes such as amylase, lipase, and trypsin but was unaffected after compensation [9]. Skeletal muscle is the largest tissue in the body, weighing up to 40% of the body. After birth, the number of muscle fibers formed by the fusion of muscle satellite cells after activation is low, and the growth and development of skeletal muscle depend mainly on the increase in muscle fiber diameter [10]. Studies have shown that there is a direct correlation between the cross-sectional area of muscle fibers and the net protein content of the muscle tissue. Muscle satellite cells are involved in muscle-fiber hypertrophy by increasing the nucleus of the muscle-fiber cell to maintain the balance between the nucleus and the cytoplasm of the myocyte [11]. Studies have demonstrated an enhanced expression of IGFI in the exercised muscle, which further activated the PI3K/Akt pathway to promote muscle-fiber hypertrophy [12] and was validated in transgenic mice overexpressing IGF1. 

Therefore, this study conducted a detailed investigation on the effects of different levels of feeding restriction on the growth performance, intestinal immunity, and skeletal muscle development of meat rabbits. Additionally, regarding whether complete compensatory growth could be achieved post 2 weeks of restriction, we analyzed the effects of feeding restriction on growth performance, intestinal immunity, and skeletal muscle development of rabbits to provide some scientific basis for the selection of feeding restriction mode for rabbits.

## 2. Material and Methods

### 2.1. Animals and Experimental Design

The test animals were obtained from Xuzhou Meat Rabbit Ecological Science and Technology Park, Jiangsu Province. Of the 35-day-old weanling rabbits of similar weight and in good health, 198 were selected. The animals were randomly divided into three groups: the control (free range), the restricted-feeding group I (fed 85% of the control), and the restricted-feeding group Ⅱ (fed 70% of the control), with 66 animals in each group, half being male and half being female. A pre-test was carried out to calculate the amount of food intake by 12 ad libitum rabbits prior to initiation of the trial period. This provided the amount of food consumption given to the three groups during the restricted feeding period. The trial was divided into 21 days of a restricted feeding period and 14 days of a compensation period.

All the rabbit hutches and equipment were strictly cleaned and disinfected by high-temperature flame a day before the trial. All test rabbits were labeled and kept in single cages in the housing. The usual feeding management and immunization procedures were used during the test period, and the environmental conditions were kept consistent. The rabbits were fed twice a day at 08:00 and 18:00, with free access to water (the diet formula and nutritional levels shown in Table 1). The food intake of growing rabbits was recorded daily. The average daily weight gain was calculated by weighing the rabbits before the start of the trial, at 56 days of age and before slaughter, and the disease and mortality of the rabbits during the trial were recorded. Six rabbits were randomly selected from each group at 56 and 70 days of age. Samples were collected from skeletal-muscle-tissue samples and three sections of small-intestine tissue.

### 2.2. Growth Performance Measurement and Health-Risk Assessment Indicators

The growth indicators such as average daily gain (ADG), average daily feed intake (ADFI), and feed-to-weight ratio (F/G) were measured as follow:Average daily gain (ADG): All test rabbits were weighed empty stomach to calculate the average daily gain of meat rabbits from 36–56 days of age (restricted feeding period), 57–70 days of age (compensation period), and 36–70 days of age (full period).Average daily feed intake (ADFI): The daily feed intake and the number of leftovers were recorded for the test rabbits for the restricted feeding period, the compensation period, and the whole period for the meat rabbits.Feed-to weight ratio (F/G): The ratio of average feed intake to average daily weight gain is the average feed-to-weight ratio.

The morbidity, mortality, and health-risk indices were calculated for the restricted feeding period, the compensation period, and the full period. Mortality due to morbidity was counted as death only.

Morbidity rate = number of rabbits with disease/total number of rabbits × 100%Mortality rate = number of rabbits killed/total number of rabbits × 100%Health-risk index = morbidity + mortality

### 2.3. Measurement of Digestive Enzyme Activity and Immune and Antioxidant Indexes

In this test, the activity of jejunal amylase, lipase, chymotrypsin, and alkaline phosphatase (ALP/AKP) was determined using the α-AMS, lipase(LPS), chymotrypsin, and alkaline phosphatase kits (all from Nanjing Jiancheng, Nanjing, China), respectively. The levels of jejunal immunoglobulin and ileal inflammatory factors were measured using ELISA kits (Shanghai Enzyme Link, Shanghai, China). The jejunal malondialdehyde (MDA) content, total antioxidant capacity (T-AOC), and superoxide dismutase (T-SOD) activities were determined using kits (Nanjing Jiancheng Company).

### 2.4. Tissue Sample Collection

Six test rabbits were randomly selected from each of the control group, restricted-feeding group I, and restricted-feeding group II at 56 days of age. One part was rinsed with PBS and fixed in 4% paraformaldehyde solution for tissue sectioning. One part was washed with PBS several times to remove blood stains on the surface, then it was placed in lyophilization tubes and frozen in liquid nitrogen for extraction of total RNA.

### 2.5. HE Staining and Muscle Fibre Size Analysis

Tissues were fixed in 4% paraformaldehyde solution, trimmed to size, and then dehydrated in different concentrations of gradient ethanol and xylene before being waxed and embedded in an embedding machine and sliced using a microtome to a thickness of 7 µm. The sections were then dewaxed and rehydrated before being stained with hematoxylin-eosin (HE), sealed with neutral gum, and photographed with a magnification of 100×. The muscle cross-sectional area was analyzed using Image 6.0.

### 2.6. RNA Extraction and cDNA Synthesis

RNA was extracted from approximately 100 mg of skeletal muscle tissue using Trizol reagent (Invitrogen-Life Technologies, Carlsbad, CA, USA) according to the manufacturer’s protocol. The concentration of extracted RNA was measured spectrophotometrically using a NanoDrop ND-1000 micro-UV-Vis spectrophotometer (NanoDrop Technologies Inc., Wilmington, DE, USA), and the purity of RNA was assessed by the A260/A280 ratio in the range of 1.8-2.0. 

Total RNA (200 μg) was reverse transcribed into cDNA using HiScript^®^ Ⅱ Q Select RT SuperMix for quantitative real-time PCR (qPCR) (Nanjing Novozymes Biotechnology Co., Ltd., Nanjing, China) according to the manufacturer’s protocol. Samples were stored at −20 °C for subsequent analysis.

### 2.7. Real-Time qPCR

Reaction system: 2 × ChamQ SYBR qPCR Master Mix 10 μL, 0.4 μL of each primer, 50×ROX Reference Dye2 0.4 μL, 1 μL of template DNA, ddH_2_O to make up to 20 μL. Amplification procedure (three-step): 95 °C pre-denaturation for 30 s; 40 cycles of reaction: 95 °C for 10 S, 60 °C for 30 s; melting curve: 95 °C 15s, 60 °C 60s, 95 °C 15 S. Primer sequences are detailed in Table 2.

### 2.8. Western Blot Analysis

The total protein was extracted by adding a certain percentage of protease inhibitor to Western and IP cell lysate and stored at −80 °C. The total protein content in skeletal muscle cells was measured using the BCA protein-concentration assay kit (Beyonce Biotechnology, Shanghai, China). The expression levels of Akt, p-Akt, p-70, and p-p70 proteins were measured using the Wes system Protein simple technique. The protein samples, the diluted primary anti-target protein antibody, GAPDH as an internal reference antibody (Proteintech, Wuhan, China), and other relevant test reagents were prepared followed by samples loading according to the Wes user procedure. Unlike traditional Western Blot, fully automated Wes gum preparation and running, membrane transfer, manual incubation or washing, press development, and subjective data processing are all done automatically in the capillary.

### 2.9. Statistical Analysis

The data were collated and counted using Excel software, and the relative gene expression was calculated using the 2^−ΔΔct^ method. Data were analyzed using SPSS 22.0 statistical software and were analyzed by one-way ANOVA and paired t-test according to different trials. The results of the gender-difference analysis showed no significant gender effect, and the results of the trials were expressed as mean ± standard deviation; *p* < 0.05 indicated a significant difference; and *p* < 0.01 indicated an extremely significant difference.

## 3. Results

### 3.1. Effects of Restricted Feeding and Compensation on Growth Performance and Health Status of Rabbits

Effects of feed restriction and compensation on growth performance of rabbits are outlined in Table 3. There was no significant difference in body weight between the restricted-feeding group I and the control group at the end of both the restricted feeding trial and the compensation trial (*p* > 0.05). Contrastingly, post 21 days of restricted feeding, the weight of rabbits in the restricted-feeding group Ⅱ decreased substantially (*p* < 0.05) compared with the control group. During the restricted feeding period, the feed-to-weight ratio of the restricted-feeding group Ⅰ and group II were significantly lower than the control group (*p* < 0.05). In the compensation stage, the average daily weight gain and feed-to-weight ratio of restricted-feeding groups I and Ⅱ demonstrated no change from the control group. The meat rabbits with a 30% restricted feeding level exhibited a significant decrease in their body weight, which was not completely compensated post 14 days of nutritional recovery. 

The effects of feed restriction and compensation on the health status of the rabbits are shown in Table 4. Compared to the control group, the health-risk index of group I and group II decreased significantly in the period of feed restriction, the compensation period, and the whole trial period. This indicates that restricted feeding promotes the lowering of the morbidity and mortality rate of growing meat rabbits.

### 3.2. Effect of Restricted Feeding and Compensation on Intestinal Alkaline Phosphatase Activity and Digestive Enzyme Activity in Rabbits

As can be seen from Table 5, the activity of alkaline phosphatase in the jejunum of broiler rabbits in the restricted-feeding group I was significantly higher than that in the control and restricted-feeding group II during the restricted-feeding phase (*p* < 0.05). There was no significant difference in alkaline phosphatase activity between the groups during the compensation period (*p* > 0.05). The effect of different levels of restriction on the activity of digestive enzymes in the jejunum of rabbits is shown in Table 6. The activity of chymotrypsin in the jejunum of rabbits in the restriction groups was significantly higher than the control group (*p* < 0.05). However, the activity of chymotrypsin in the restriction group II compared with the restriction group I was higher (*p* < 0.05).

### 3.3. The Effect of Restricted Feeding and Compensation on the Immune Index and Antioxidant Capacity of the Rabbit Intestine

The effects of feed restriction and compensation on immune indices in jejunum of rabbits are outlined in Table 7. The secretory immunoglobulin A (sIgA) level was significantly increased and decreased in the restricted-feeding group I and group II, respectively, compared to the control group. On the other hand, immunoglobulin G (IgG) levels in the restricted-feeding group II were significantly lower than the control group and the restricted-feeding group I (*p* < 0.05), indicating that restricted feeding could improve the immune function of the rabbits to some extent. The levels of various immunoglobins such as sIgA, IgG, and immunoglobulin M (IgM) in the restricted-feeding groups I and II were unaltered at the end of the 14-day compensation period (*p* > 0.05).

The effects of feed restriction and compensation on antioxidant capacity in the jejunum of rabbits are shown in Table 8. During the restricted-feeding period, the activity of T-AOC (superoxide dismutase) and T-SOD (total antioxidant capacity) in group II were significantly lower, while the MDA content of rabbits in groups I and II both showed a significant decrease (*p* < 0.05). The activity of T-SOD was significantly restored post 14 days of compensatory growth of restricted-feeding group II than the control group and the restricted-feeding group I (*p* < 0.05), whereas the MDA content remained low in restricted-feeding groups I and II (*p* < 0.05).

As can be seen from Table 9, inflammatory marker interleukin-10 (IL-10) levels in restricted-feeding groups I and II were significantly higher compared to the control group at 56 days of age (*p* < 0.05). At 70 days of age, the interferon-γ (IFN-γ)and tumor necrosis factor-α (TNF-α) levels in restricted-feeding group II were significantly higher than the control group (*p* < 0.05).

### 3.4. Effects of Restricted Feeding and Compensation on Skeletal Muscle Development and Related Gene Expression in Rabbits

Hematoxylin-eosin (HE) staining confirmed that the skeletal muscle fibers in the control group were intact with uniform muscle gaps. Compared with the control group, the area of skeletal muscle fibers was lower in the restricted-feeding groups I and Ⅱ and (*p* < 0.05) (Figure 1A, Table 10). The mRNA expression of IGF1, IGF2, MSTN, and FOXO1 were quantitated of three groups where IGF1 mRNA expression was unaffected in group I, whereas group II showed a dramatic reduction (*p <* 0.05). Additionally, MSTN and FOXO1 transcript levels were upregulated in restricted group II compared to the control group (Figure 1B).

Further, the total Akt proteins levels were unaltered in group II, whereas the p-Akt level reduced significantly (Figure 2, Appendix A). Similarly, the downstream effector molecule of Akt, p70, behaved as Akt in a 30% restricted-feeding group II compared to the control group. Restriction feeding reduced the protein levels of total Akt and p-Akt like AMPK.

## 4. Discussion

### 4.1. Restricted Feeding Influences the Growth Performance and Digestive Physiology of Meat Rabbits

Nutrition consumption is an important factor in the early growth and development of rabbits where body weight is a visual indicator. It was observed that during the restriction period, the final body weight of group II was significantly lower than the control group, and both group I and group II showed reduced feed-to-weight ratios compared to the control group. This indicates that restricted feeding of rabbits promotes an enhanced feed-conversion rate consistent with the findings of Oliveria [13] and Abou-Kassem et al. [14]. However, no major changes were observed in growth post 14 days of the compensatory growth period in restricted-feeding group II, which could possibly be due to the differences in the intensity and duration of restricted feeding and the duration of compensatory growth [15]. Moreover, nutritional limitations have various advantages such as a reduction in the diarrhea rate and the mortality of growing meat rabbits and improving the survival rate of litters, especially weanlings. These findings evidently highlight the benefits of restricted feeding in meat rabbits on their survival rate.

The efficiency activity of digestive enzyme activity in the intestine is the reflection of digestion and absorption capacity [16]. Therefore, we noticed the enhanced chymotrypsin activity, thereby promoting the digestion and absorption of protein in restricted-feeding groups. Similarly, alkaline phosphatase, a marker enzyme for the epithelial cells of the small intestinal villi, ameliorated the damage to the intestinal mucosa caused by metabolic waste such as endotoxin [17]. The restricted-feeding group I showed significantly higher alkaline phosphatase activity compared to the control group and the restricted-feeding group II. This indicates that a certain degree of restricted feeding can promote the digestive and absorption function of the intestine and improve the function of the intestinal immune barrier.

### 4.2. Restricted Feeding Influences the Immune Function and Antioxidant Capacity of Meat Rabbits

The intestine is an important organ for keeping animals healthy. In addition to absorbing nutrients, the intestine is also a barrier against harmful substances and bacterial invasion. The intestinal immune system also plays an important role in this process [18]. sIgA is the main effector of mucosal immunity with the capability to block the invasion of toxic microorganisms [19]. We figured out that rabbits with a 15% restriction demonstrated high sIgA compared with the control group, which signifies enhanced immunity during the restriction period. Dissimilarly, the 30% restriction group exhibited a significant decrease in sIgA and IgG protein levels, possibly due to the lack of certain amino acids caused by the reduced protein intake and the blocked synthesis of immune proteins in the intestinal mucosa, which in turn affected the immune function of the intestine. 

Interleukins are a group of cytokines with multiple biological roles [20]. Among them, IL-1β stimulates the production of inflammatory mediators through autocrine or paracrine secretion and aggravates immune-mediated tissue damage [21]. IFN-γ and TNF-αon intestinal epithelial cells increase mucosal tissue permeability [22]. This study claims an enhanced level of IL-10 (anti-inflammatory factor) in the ileal mucosa in the restricted-feeding group, thus reducing the occurrence of diarrhea and promoting the healthy growth of rabbits.

The total antioxidant capacity can reflect the body’s level of compensation in response to the external environmental factors and the metabolism of free radicals in the body [23]. It is measured through the level of malondialdehyde (MDA), which directly reflects the degree of peroxidation in the body [24]. The restricted group manifested a considerable decrease in MDA content implying reduced lipid peroxidation in rabbits and increased antioxidant enzyme activity.

### 4.3. Different Levels of Restriction Have an Effect on Skeletal Muscle Development in Rabbits

The maintenance of skeletal muscle is the manifestation of the normal physiological function of the body. Skeletal muscle development profoundly decreased in the 30% restricted-feeding group with a decrease in the area of skeletal muscle fibers compared to the control group. IGF1 and IGF2 are vital effector molecules in skeletal muscle development, repair and regeneration, and hypertrophy of muscle fibers in animals [25]. We observed the negative influence of nutritional limitation on IGF1-mediated skeletal muscle development. Therefore, it concludes that IGF1 expression is determined by nutritional status [26].

Muscle growth and development is regulated by IGF1-mediated PI3K/Akt pathway [27]. mTOR is activated through various metabolites and regulated by AKT signaling [28]. This suggests that 30% restriction of feeding levels may cause inhibition of mTOR activity, which causes downregulation of its downstream proteins.

Several studies have shown that the expression of MSTN mRNA is influenced by nutritional status, breed, and developmental stage [29]. Restriction-feeding group II exhibited a substantial increase in MSTN expression content in muscle tissue. The forkhead transcription factor FOXO1 is specifically expressed in muscle tissue and possess an important role in muscle formation and Akt dependent sarcoplasmic protein synthesis [30]. Presently, only 30% of the restriction level changed the expression level of the FOXO1 gene in the skeletal muscle of rabbits, which may be related to the intensity of restriction and the size of rabbits.

## 5. Conclusions

In summary, the 15% restriction level is suitable for the growth and development of the rabbit, ensuring normal development of the animal body while reducing diarrhea and mortality. The 30% feeding level reduced the growth rate of the meat rabbits with no compensatory growth post 2 weeks of nutritional recovery. In addition, due to the reduction in the body’s nutritional level, the synthesis of immunoglobulin and antioxidant enzymes were blocked, and the level of pro-inflammatory factors during the compensation period had increased. The 30% restriction level affected the skeletal muscle growth and development in growing rabbits by regulating the PI3K/Akt signaling pathway. 

## Figures and Tables

**Figure 1 animals-12-00160-f001:**
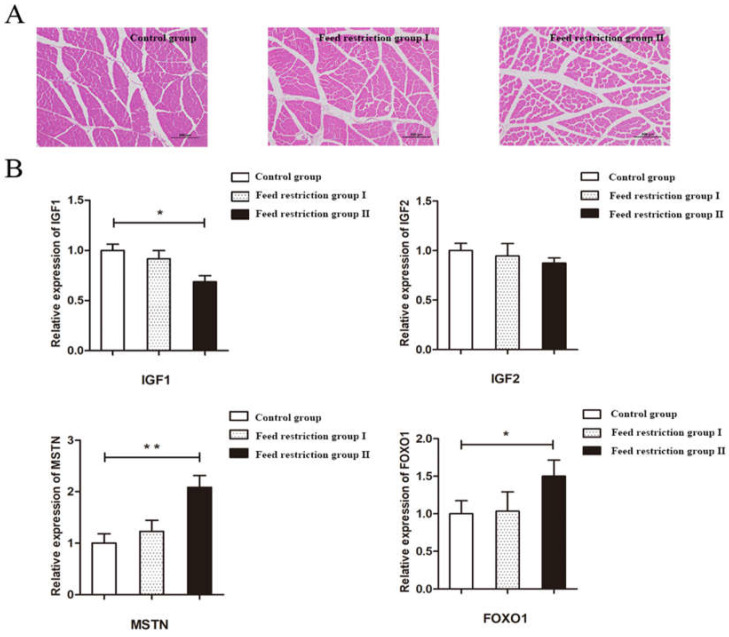
The HE staining image and effect of feed restriction associated with skeletal muscle development. (**A**) the HE staining image of the skeletal muscle of the control group, the restricted-feeding group I, and the restricted-feeding group Ⅱ, respectively; (**B**) effect of feed restriction on the expression level of genes associated with skeletal muscle development; * indicated significant difference (*p* < 0.05); ** indicated extremely significant difference (*p* < 0.01).

**Figure 2 animals-12-00160-f002:**
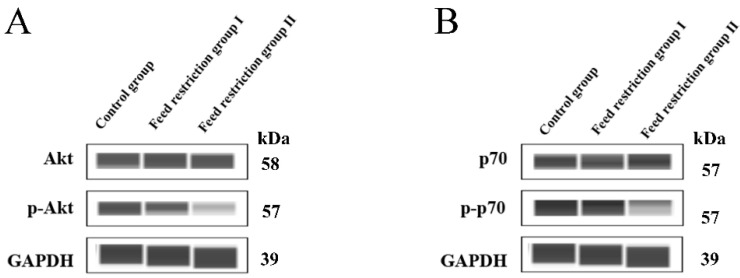
Effect of feed restriction on the phosphorylation level of Akt and p70. (**A**) the Akt, p-Akt, and GAPDH protein expressions in different groups; (**B**) the p70, p-p70, and GAPDH protein expressions in different groups.

**Table 1 animals-12-00160-t001:** Composition and nutrient levels of the basal diet.

Ingredients	Content, %	Nutrient levels	Content
Corn	5.70	DM, %	89.16
Wheat	8.00	DE, MJ/kg	2.52
Soybean meal	9.00	CP, %	16.09
Sesame meal	4.30	EE, %	3.30
Rapeseed meal	2.30	CF, %	18.00
Corn germ meal	13.00	Ash, %	9.45
Sprouting corn bran	3.30	Ca, %	1.00
Wheat bran	14.30	P, %	0.66
Rice husk powder	12.00	NDF, %	35.04
Sorghum shell	3.70	ADF, %	21.11
Artemisia argyi powder	4.70	Lignin, %	2.57
Peanut vine powder	7.00	Lys, %	0.84
Garlic skin	8.00	Met, %	0.33
Corn oil	0.70	A SAA, %	0.57
Premix	4.00	Arg, %	0.85
Total	100.00	Thr, %	0.50
		ST, %	13.99
		NaCl, %	0.36
		Vit. A, IU/kg	8000.00
		Vit. D, IU/kg	800.00
		Vit. E, IU/kg	40.00

**Table 2 animals-12-00160-t002:** Primers for qPCR.

Gene Names	Primer sequences(5′→3′)
*IGF1*	F: TCTGAGGAGGCTGGAGATGT
	R: TGTTGGTAGATGGAGGCTGA
*IGF2*	F: TGGCATTGTGGAGGAATGCT
	R: ACTTGCCCACGGAGTAATCG
*MSTN*	F: CGCCTGGAAACAGCTCCTAA
	R: TTGTTTCCGTCGTAGCGTGA
*FOXO1*	F: CATGCTACTCGTTTGCACCG
	R: TTTGCATAGGCATCTGGGGC
*GAPDH*	F:CACCAGGGCTGCTTTTAACTCT
	R:CTTCCCGTTCTCAGCCTTGACC

**Table 3 animals-12-00160-t003:** Effects of feed restriction and compensation on growth performance of rabbits.

Items		Groups		
		Control Group	Group I	Group II
35 d starting weight, g		886.36 ± 23.73	886.36 ± 25.42	886.06 ± 28.51
56 d final weight, g		1527.30 ± 53.71 ^a^	1501.96 ± 38.97 ^a^	1405.93 ± 24.90 ^b^
70 d final weight, g		1982.09 ± 84.75 ^a^	1968.04 ± 52.53 ^a^	1841.36 ± 52.01 ^b^
36–56 d	ADG, g	30.51 ± 4.64 ^a^	29.31 ± 2.45 ^a^	24.69 ± 4.36 ^b^
	ADFI, g	87.80 ± 6.45	74.73 ± 5.47	65.83 ± 6.63
	F/G	2.87 ± 0.25 ^a^	2.55 ± 0.18 ^b^	2.48 ± 0.21 ^b^
57–70 d	ADG, g	32.52 ± 4.33	33.28 ± 3.36	31.07 ± 3.56
	ADFI, g	104.29 ± 8.33	108.08 ± 9.52	94.24 ± 7.18
	F/G	3.22 ± 0.38	3.24 ± 0.49	3.17 ± 0.64
36–70 d	ADG, g	31.49 ± 3.73	31.30 ± 2.47	27.89 ± 3.78
	ADFI, g	95.91 ± 7.64 ^a^	88.47 ± 6.28 ^a^	76.51 ± 7.28 ^b^
	F/G	3.06 ± 0.20	2.90 ± 0.16	2.83 ± 0.34

Note: Different letters on the same line mean significant difference (*p* < 0.05); the same letters indicate insignificant differences (*p* > 0.05). “d” means day-old.

**Table 4 animals-12-00160-t004:** Effects of feed restriction and compensation on health status of rabbits (%).

Times	Index	Groups		
		Control Group	Group I	Group II
36–56 d	Total number of rabbits	66	66	66
	Morbidity	6.06	4.55	3.03
	Mortality	9.09	3.03	3.03
	Health-risk evaluation	15.15	9.09	6.06
57–70 d	Total number of rabbits	60	64	64
	Morbidity	6.67	3.12	6.25
	Mortality	9.84	6.25	9.38
	Health-risk evaluation	18.33	9.38	15.63
36–70 d	Total number of rabbits	66	66	66
	Morbidity	12.12	7.58	9.09
	Mortality	18.18	9.09	12.12
	Health-risk evaluation	30.30	16.67	21.21

**Table 5 animals-12-00160-t005:** Effects of feed restriction and compensation on alkaline phosphatase activity in rabbits.

Times	Index	Groups		
		Control Group	Group I	Group II
56 d	Alkaline phosphatase activity (King unit/mgprot)	168.25 ± 32.94 ^b^	192.53 ± 27.25 ^a^	155.80 ± 29.99 ^b^
70 d		123.96 ± 24.07	134.40 ± 12.68	129.44 ± 28.07

Note: Different letters on the same line mean significant difference (*p* < 0.05); the same letters indicate insignificant differences (*p* > 0.05).

**Table 6 animals-12-00160-t006:** Effects of feed restriction and compensation on digestive enzyme activities in rabbits (U/gprot).

Times	Index	Groups		
		Control Group	Group I	Group II
56 d	Chymotrypsin	66.69 ± 6.25 ^c^	74.77 ± 5.26 ^b^	86.63 ± 8.32 ^a^
	Amylase	2.25 ± 0.18	2.34 ± 0.21	2.11 ± 0.24
	Lipase	44.94 ± 3.17	48.90 ± 4.98	40.95 ± 6.60
70 d	Chymotrypsin	78.73 ± 8.42	80.74 ± 7.15	75.89 ± 5.92
	Amylase	1.74 ± 0.23	1.81 ± 0.14	1.78 ± 0.13
	Lipase	34.78 ± 4.05	37.49 ± 4.27	36.48 ± 4.75

Note: Different letters on the same line mean significant difference (*p* < 0.05); the same letters indicate insignificant differences (*p* > 0.05).

**Table 7 animals-12-00160-t007:** Effects of feed restriction and compensation on immune indexes in rabbits (mg/g).

Times	Index	Groups		
		Control Group	Group I	Group II
56 d	sIgA	10.47 ± 1.11 ^b^	11.92 ± 1.03 ^a^	9.07 ± 0.94 ^c^
	IgG	29.24 ± 2.68 ^a^	29.83 ± 3.36 ^a^	25.33 ± 3.38 ^b^
	IgM	17.61 ± 2.90	17.17 ± 1.97	17.57 ± 2.27
70 d	sIgA	8.84 ± 0.89	8.81 ± 0.85	8.76 ± 0.47
	IgG	27.02 ± 2.39	26.72 ± 2.91	27.25 ± 2.54
	IgM	15.65 ± 1.55	15.18 ± 1.35	15.40 ± 1.60

Note: Different letters on the same line mean significant difference (*p* < 0.05); the same letters indicate insignificant differences (*p* > 0.05).

**Table 8 animals-12-00160-t008:** Effects of feed restriction and compensation on antioxidant capacity in jejunum of rabbits.

Times	Index	Groups		
		Control Group	Group I	Group II
56 d	T-AOC, mmol/g	0.88 ± 0.12 ^a^	0.84 ± 0.11 ^a^	0.70 ± 0.16 ^b^
	T-SOD, U/mgprot	178.02 ± 25.17 ^a^	181.00 ± 25.93 ^a^	143.05 ± 23.01 ^b^
	MDA, nmol/mgprot	1.87 ± 0.36 ^a^	0.86 ± 0.18 ^b^	0.89 ± 0.26 ^b^
70 d	T-AOC, mmol/g	0.98 ± 0.15	0.96 ± 0.13	0.93 ± 0.21
	T-SOD, U/mgprot	143.99 ± 25.89 ^b^	153.19 ± 33.53 ^b^	181.37 ± 25.84 ^a^
	MDA, nmol/mgprot	2.40 ± 0.55 ^a^	1.52 ± 0.31 ^b^	1.62 ± 0.65 ^b^

Note: Different letters on the same line mean significant difference (*p* < 0.05); the same letters indicate insignificant differences (*p* > 0.05).

**Table 9 animals-12-00160-t009:** Effects of feed restriction and compensation on inflammatory factors in rabbits (ng/gprot).

Times	Index	Groups		
		Control Group	Group I	Group II
56 d	IL-10	159.03 ± 10.23 ^b^	171.82 ± 6.12 ^a^	176.90 ± 10.94 ^a^
	IL-1β	32.92 ± 3.53	34.05 ± 3.02	32.16 ± 1.63
	IFN-γ	152.78 ± 8.73	152.71 ± 5.98	154.75 ± 6.92
	TNF-α	90.18 ± 7.92	87.03 ± 3.50	93.74 ± 6.53
70 d	IL-10	155.47 ± 7.72	156.25 ± 12.77	159.98 ± 8.71
	IL-1β	29.12 ± 1.97	33.05 ± 5.29	34.60 ± 5.25
	IFN-γ	146.55 ± 13.08 ^b^	154.11 ± 19.68 ^b^	175.99 ± 10.68 ^a^
	TNF-α	95.17 ± 10.58 ^b^	97.02 ± 12.53 ^b^	114.28 ± 15.95 ^a^

Note: Different letters on the same line mean significant difference (*p* < 0.05); the same letters indicate insignificant differences (*p* > 0.05).

**Table 10 animals-12-00160-t010:** ANOVA of the area of the skeletal muscle fibers.

Index	Group		
	Control group	Group I	Group II
Area of the skeletal muscle fibers (μm²)	1147.44 ± 98.94 ^a^	956.62 ± 82.77 ^b^	927.41 ± 60.17 ^b^

Note: Different letters on the same line mean significant difference (*p* < 0.05); the same letters indicate insignificant dif-ferences (*p* > 0.05).

## Data Availability

The datasets generated for this study are available upon request to the corresponding author.

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
