# Peer review of "Effects of Restricted Feeding on Growth Performance, Intestinal Immunity, and Skeletal Muscle Development in New Zealand Rabbits"

_animals, 2022, doi:10.3390/ani12020160_

Round 1

Reviewer 1 Report

In this manuscript “Effects of restricted feeding on growth performance, intestinal immunity and skeletal muscle development in New Zealand rabbits”, authors showed that the 15% feeding limit unaffected the weight and skeletal muscle development of rabbits whereas the 30% feeding limit affected them in growing rabbits. Feed restriction could influence the phosphorylation level of Akt and p70 via activating PI3K/Akt signaling pathway in skeletal muscle. Those fingdings are interesting. A few minor revisions are list below.

  1. In the introduction, please refer to the related research on rabbits to illustrate the selection basis of 15% and 30% feeding restriction.
  2. 2.Why only consider intestinal immunity,but not serum immunity? Please add corresponding explanations in the discussion.
  3. The results of western blot seem to be an electronic image, which is different from the result of traditional WB? How is the ratio of the histogram obtained? It should be stated in the method. At the same time, the size of the protein should be indicated on the figure.
  4. 4. The format needs further modification. For example, in the table, “I group” and “II group” should be “Group I” and “Group II”. L240 should not be bolded.
  5.  list the  calculation methods of  morbidity, mortality and health risk indices in  Material and methods 
  6.  the bolds of AKT, P-AKE, P70 and p-p70 should be original and no clips, please replace them

Reviewer 2 Report

Introduction

 bibliography on other animal species is not necessary. On the rabbit there are several researches: for example I point out Gidenne et al. 2003 and Bovera et al. 2008 (Gidenne, T.; Feugieri, A.; Jehl, N.; Arveux, P.; Boisot, P.; Briens, C.; Corrent, E.; Fortune, H.; Montessuy, S.;Verdelhan, S. Un rationnement alimentaire quantitatif post-sevrage permet de réduire la fréquence des diarrhées, sans dégradation importante des performances de croissance : résultats d'une étude multi-site. 10èmes Journées de la Recherche Cunicole, 19-20 nov. 2003, Paris, 29-32.) (Bovera, F.; Di Meo, C.; Marono, S.; Vella, N.; Nizza, A. Feed restriction durin summer: effect on rabbit grouth performance. 9th World Rabbit Congress – June 10-13, 2008 – Verona – Italy 567-572)

 Line 47 change “free-range” as “ ad libitum”

Lines 49-50 delete “There have been studies reporting a low mortality rate when restricted feeding is employed during broiler rearing3

Line 90 and 92 change “free-range” as “ ad libitum”

Table 1 change “Lingin” as “Lignin”

 Table 1 change   “ VA, VD, VE,” as “ Vit. A, Vit. D,Vit. E”

Lines 172-173 change “There was no significant difference in body weight between the restricted feeding group and the control group” as  “There was no significant difference in body weight between the restricted feeding group I and the control group”   

Line 183 change “Group I” as “group I”   

Table 4 report letters for significant differences

Line 195 change   “feeding group” as “feeding group II”

Lines 200-201 change      “However, the activity of chymotrypsin in the restriction group II

compared with the restriction group I was unaltered” as “However, the activity of chymotrypsin in the restriction group II compared with the restriction group I was higher”

Line 259 change “Oliveria 14.” As “Oliveria 14 and Abou-Kassem et al. 15.”

Abou-Kassem, D.E.;  Mahrose, K.M.; El-Samahy, R.A.; Shafi, M.E.; El-Saadony, M.T.; El-Hack, M.E.A.; Emam, M.; El-Sharnouby, M.; Taha, A.E.; Ashour, E.A. Influences of dietary herbal blend and feed restriction on growth, carcass characteristics and gut microbiota of growing rabbits. It. J. Anim. Sci. 2021, 20, 896–910.
